# Long-Term Comparison of Two- and Three-Dimensional Computed Tomography Analyses of Cranial Bone Defects in Severe Parietal Thinning

**DOI:** 10.3390/diagnostics14040446

**Published:** 2024-02-17

**Authors:** Johannes Dominikus Pallua, Anton Kasper Pallua, Werner Streif, Harald Spiegl, Clemens Halder, Rohit Arora, Michael Schirmer

**Affiliations:** 1Department of Orthopaedics and Traumatology, Medical University of Innsbruck, Anichstraße 35, 6020 Innsbruck, Austria; rohit.arora@i-med.ac.at; 2Former Institute for Computed Tomography-Neuro CT, Medical University of Innsbruck, Anichstraße 35, 6020 Innsbruck, Austria; anton.k.pallua@cnh.at; 3Department of Pediatrics I, Medical University of Innsbruck, Anichstrasse 35, 6020 Innsbruck, Austria; werner.streif@i-med.ac.at; 4WESTCAM Datentechnik GmbH, Gewerbepark 38, 6068 Mils, Austria; harald.spiegl@westcam.at (H.S.); clemens.halder@westcam.at (C.H.); 5Department of Internal Medicine, Clinic II, Medical University of Innsbruck, 6020 Innsbruck, Austria

**Keywords:** parietal thinning, measurement reliability, bone defects, magnetic resonance imaging, computer tomography, osteology, rheumatology, rare disease

## Abstract

Parietal thinning was detected in a 72-year-old with recurrent headaches. Quantification of bone loss was performed applying two- and three-dimensional methods using computerized tomographies. Two-dimensional methods provided accurate measurements using single-line analyses of bone thicknesses (2.13 to 1.65 and 1.86 mm on the left and 4.44 to 3.08 and 4.20 mm on the right side), single-point analyses of bone intensities (693 to 375 and 403 on the left and 513 to 393 and 411 Houndsfield Units on the right side) and particle-size analyses of low density areas (16 to 22 and 12 on the left and 18 to 23 and 14 on the right side). Deteriorations between days 0 and 220 followed by bone stability on day 275 were paralleled using the changed volumes of bone defects to 1200 and finally 1133 mm^3^ on the left side and to 331 and finally 331 mm^3^ on the right side. Interfolding as measurement of the bones’ shape provided changes to −1.23 and −1.72 mm on the left and to −1.42 and −1.30 mm on the right side. These techniques suggest a stabilizing effect of corticosteroids between days 220 and 275. Reconstruction of computerized tomographies appears justified to allow for quantification of bone loss during long-term follow-up.

## 1. Introduction

Biparietal thinning, also known as biparietal osteodystrophy, is characterized by extreme bone thinning in the parietal bones, located between the sagittal suture and parietal prominence and anteriorly to the parietal foramina [1,2]. This condition is usually expressed symmetrically and bilaterally and can be classified into a simple flat type and a groove or crater-like type [1,2]. The prevalence of all types of biparietal thinning is between 0.25% and 0.8% of the population [1,3]. Biparietal thinning is common in older (postmenopausal) women, although it may also occur in men and children [1,4,5]. This aspect differs from Paget’s disease, which shows a male–female ratio of 3:2 when using sex-standardized prevalence rates [6].

Parietal skull thinning is a complex phenomenon involving bone density, structure, and composition changes. Although first described in 1783, the etiology of biparietal thinning remains unknown [1]. Several etiologic factors have been proposed, including age-related bone loss, hormonal changes, genetic predisposition, and nutritional deficiencies. In addition, medical conditions like osteoporosis, Paget’s disease, and hyperparathyroidism have been associated with parietal skull thinning, as well as syphilis, tumors, external pressure, muscular stimuli, and friction of the galea aponeurotica [2,7,8,9]. Tumors, for example, can cause bone loss through direct invasion or indirectly by releasing cytokines that stimulate bone resorption. Some cases of parietal thinning have been observed with suspected genetic backgrounds in families, but the role of genetics has not been established so far [4,10]. In summary, biparietal thinning is a pathological lesion of decreased bone density and quality, which increases the risk of skull fractures.

For diagnostic purposes, radiographic imaging including conventional X-ray, computed tomography (CT), and magnetic resonance imaging (MRI) has been used together with bone scintigraphy [3,4,11,12]. These techniques can provide detailed structural information about the thickness and density of the parietal bone with CT and MRI and signs of metabolic activity when using bone scintigraphy. Conventional radiography has limited sensitivity for the early detection of bone loss and cannot provide quantitative data on bone density. More sensitive imaging modalities to detect bone loss include CT and MRI. While CT is ideal for evaluating the extent and severity of bone loss, MRI is better suited for evaluating soft tissues and bone marrow, but it has limited spatial resolution and may not provide sufficient details for assessing bone density [13]. Quantitative measures of bone density, such as bone mineral density (BMD) using dual-energy X-ray absorptiometry (DXA) or other specialized techniques, are certainly not sensitive enough for the skull, although considered as the gold standard for diagnosing osteoporosis in other bone regions [14].

Diagnosis of parietal skull thinning typically requires imaging studies with quantitative measures of bone density. The present investigation aims to retrospectively analyze available images with quantification of chronic parietal skull thinning during follow-up. The novelty of this study lies in its focus on applying advanced imaging technologies to compare the descriptive approach to CT images with two-dimensional (2D) and three-dimensional (3D) quantitative techniques before and after corticosteroid treatment.

## 2. Materials and Methods

### 2.1. Computed Tomography Imaging

CT images were obtained with a CT scanner (SIEMENS SOMATOM Confidence; Siemens, Erlangen, Germany). The patient’s head was scanned with a tube voltage of 120 kVp, a maximum X-ray tube current of 184 mA, an X-ray tube current of 112 mA, an exposure time per rotation of 1 s, a nominal single collimation width of 0.6 mm, a nominal total collimation width of 12 mm, and a pitch factor of 0.55.

### 2.2. Application of Quantification Methods for Image Analyses

Image analyses were performed using CT datasets from 4 different time points. The reconstructions were carried out using Analyze 14.0 (Analyze Direct Inc., Overland Park, KS, USA) and GOM Inspect (GOM, Braunschweig, Germany) software. In the first stage of the assessment, the head CT scans were reconstructed via semi-automatic object segmentation with region growth (minimum 319, maximal 3071).

#### 2.2.1. Two-Dimensional (2D) Measurement Methods

Two-dimensional measurements were based on analyzing the CT data gained with multiplanar reconstructions. The image was set in three planes—frontal, sagittal, and transverse—and measurements were performed with the sample point tool and sample line tool. The sample line tool defines a line on a 2D image or a 3D rendering. The tool reports distance measurements, a line intensity profile, and the coordinates of the line endpoints. The sample point tool allows for the selection of points on the 2D image data or 3D rendering (Figure 1). The tool reports the coordinates and intensity values of the selected point.

#### 2.2.2. Computed Tomography Osteoabsorptiometry

Computed tomography osteoabsorptiometry (CT-OAM) is a non-invasive technique that can reveal information about the density distribution. In contrast to conventional methods of CT densitometry, which measure absolute values for bone density in large areas, CT-OAM can demonstrate the relative distribution over the entire surface using a false-color diagram. This technique allows for the detection of differences in relative distribution [15,16,17,18]. CT-OAM was evaluated using Analyze 14.0 (Analyze Direct Inc., Overland Park, KS, USA). Skull CT datasets were registered with 2D transformation using linear interpolation and manually segmented before the data were false-color-coded and superimposed on a three-dimensional reconstruction for anatomical localization of the mineral bone density, creating a bone mineral density densitogram. The maximum intensity projection revealed the HU of each pixel to a depth of 190 slices (149.570 mm), and the threshold value was chosen to be ≤300 to ≥1300 HU.

#### 2.2.3. Analysis of Densitogram Patterns

The analysis of densitogram patterns was adapted from Poilliot A. et al. and Gay M.H. et al. [15,19]. The skull’s mineral density pattern over time was evaluated based on the mean HU values of the regions on the densitogram for each dataset. The skull surface was subdivided into three regions: Os parietal left and right and sutura sagittalis. Zones with low mineralization were defined as those with values of 301 to 400 HU. These lower-mineralization zones were identified by generating a black and white image with a threshold cut-off at 500 HU using Fiji (ImageJ) [20]. The regions with low HU values were identified based on their anatomical location with automatic particle counting using Fiji. The CT-OAM methodology is depicted in Figure 2.

#### 2.2.4. Three-Dimensional (3D) Measurement Method

For 3D measurements based on CT data, all scans were reconstructed in a 3D space using the 3D Slicer 5.6.1 (http://www.slicer.org, accessed on 15 January 2023). The program allowed for the conversion of a DICOM file into a Mesh file, which could be further evaluated using the GOM Inspect (GOM, Braunschweig, Germany) software. One observer conducted 3D reconstruction to avoid any errors associated with the 3D reconstruction itself. The CT scans were marked as “model” and converted into the CAD format, while the second tomogram, called “comparative,” was converted into the Mesh format, a universal file format for geometry. Both models were compared using the GOM Inspect (GOM, Braunschweig, Germany) program (Figure 3). Next, all reconstructed skulls were aligned over the red-marked area. The parietal bone thickness itself was then measured in three defined positions for every time point. Finally, the volumes of bone loss were calculated for both sides.

### 2.3. Statistical Analyses

This study was guided by an earlier study on the reliability of glenoid bone defect measurements on 2D and 3D CT [21]. Accordingly, two models were applied to test intra- and inter-observer reliability. We added an observer (an orthopedic surgeon) to present the reliability of measurements performed by observers with the same level of expertise. Thus, three independent researchers at different levels of experience were involved.

Statistical analysis was performed using GraphPad Prism version 9 (GraphPad Prism Software, Inc., San Diego, CA, USA) and Statistics 22.0 (IBM SPSS Statistics for Windows, V.21.0. IBM Corp., Armonk, NY, USA). After testing for normal distribution, linear and volumetric measurements were compared over time using the one-way ANOVA test and Fisher’s LSD test, with *p*-values < 0.05 considered as significant.

## 3. Results

### 3.1. Patient’s Assessment and Follow-Up with Multimodal Diagnostics

A 72-year-old female patient presented at the rheumatology outpatient clinic with recurrent headaches. Headaches occurred together with the development of subjective bony swellings on the skull. History revealed subtotal gastrectomy with T1 lymph node dissection because of a malignant neoplasm (stage pT1(a)N0M0; R0; G3; L0; V0, UICC Ia) three years earlier, which was in ongoing remission after the clinical screening with additional gastroscopy, the first CT, and another ^18^F-FDG-PET-CT during follow-up.

As the patient had noticed parietal changes for 1 year, ^99m^Technetium scintigraphy, MRI, and CT imaging were performed. The MR confirmed not only the irregularities of the skull but also showed chronic vascular leukencephalopathic changes. Endocrinology results were used to diagnose hyperparathyroidism (with an elevated level of 88.1 ng/L for parathormone with an upper normal level of 65 ng/L) and to rule out typical osteodystrophy deformans (Paget’s disease) based on the ^99m^Technetium scintigraphy and MRI. Osteoporosis was diagnosed in the lumbar spine with a *T* score of −2.6, and osteopenia of the femur (*T* = −1.8) using DXA (HOLOGIC), but without the need for bone-specific treatment as argued according to the current guideline of the “Dachverband Osteologie” [22]. A dermatologic visit with histology of a capillitium skin biopsy (including dermis and minimal subcutis) had shown cicatrized inflammatory reactions around the hair follicles with diffuse lymhocytic infiltrates; scarring alopecia; and Lichen ruber, planopilaris type, which was not suspected as the underlying cause of parietal thinning. A neurological visit ruled out any neurological disease as the underlying cause of parietal thinning. From a clinical perspective, the patient was otherwise healthy; they did not have any elevated markers for inflammation (erythrocyte sedimentation rate and C-reactive protein) in the serum or any vascular risk factors, including arterial hypertension, diabetes mellitus, obesity, or smoking history.

Considering the diffuse lymphocytic infiltrates as possible triggers of parietal thinning, the patient was informed about the potential benefits and risks of immunosuppression, using specific corticosteroids as a rapidly acting immune-suppressive agent. After a shared decision-making process, 40mg of methylprednisolone was rapidly tapered to 8mg, with a marked reduction in headaches both in terms of the frequency of occurrence and severity. Furthermore, biphosphonates were recommended for the treatment of osteoporosis. CT was repeated at days 185, 229, and 275 to assess further bone loss as a side effect of corticosteroids to assess the possible worsening of parietal thinning. In the CTs, the nominal single collimation width was 0.6mm and the nominal total collimation with was 12mm.

### 3.2. Descriptive Imaging Reports before and after 3D Reconstruction

Without reconstruction, the skull of the first CT was described as having asymmetric parietal thinning of the left side as a normal variant. Retrospective 3D reconstruction of the CT revealed further details of the normal variant, with impressive defects at both sites of the skull (Figure 4). These findings were confirmed in the subsequent CTs performed on days 185, 229, and 275 (Figure 5).

Next, the volume rendering and maximum intensity projection of the skull were performed in all follow-up CTs, suggesting signs of disease progression, which justified further analyses to quantify the changes over time (Figure 5).

### 3.3. Two-Dimensional (2D) Measurements of Biparietal Thinning

The parietal bone was measured at the three thinnest points on days 0, 185, 229, and 275 (Figure 6).

Then, 2D measurements were prepared in 3D head models for all four CT images with three points manually selected using the sample point tool (Figure 7a–c for each side) and the selection of the same points in the follow-up CTs. The results of sample line measurements, sample point measurements, and particle analysis are shown in Figure 7 and Figure 8, with deterioration in parietal thinning between day 0 and day 229 but without further deterioration between day 229 and 275, indicating stable disease without further bone loss despite treatment with corticosteroids. This finding paralleled the improvement of headaches under treatment with corticosteroids. Only one value was available for particle analysis.

### 3.4. Three-Dimensional (3D) Measurements of Biparietal Thinning

To further improve the quantification of bone loss over time, 3D measurements were planned using a computer-aided design (CAD) model. The preparation of the CAD model with an overlay on the Mesh format and skull alignment is shown in Section 2. With average differences between the two models not exceeding 0.15 mm, the 3D reconstruction is considered reliable.

Using this 3D model, the parietal thicknesses can be displayed as a color map painted onto the surface (Figure 8). This method assures the complete mapping of bone thinning over the skull and clearly shows the two locations on the left and the right side.

Next, interfolding and volumetric measurements were calculated for both sides, as summarized in Figure 9. For volume analysis, no *t*-test was possible since there was only one value.

### 3.5. Comparisons of Changes in Bone Loss over Time

A comparison of the results is presented in Figure 10 and Table 1, showing changes in bone loss over time when using different methods. Bone loss was significant between days 0 and 229, independent from the method applied, whereas bone loss was at least stable without significant deterioration between days 229 and 275.

## 4. Discussion

This study summarizes the fundamental methodological approaches currently available to quantify bone loss in parietal thinning for follow-up comparisons based on data from a patient with actual changes of parietal thinning over time. Using these methods, both 2D and 3D methods provided results that are proposed to be considered superior to a pure description of parietal thinning for comparative follow-up analyses. Most notably, the results show changes in parietal thinning, with significant deterioration and stable disease under clinical symptoms during follow-up.

All methods applied are based on low-dose CTs as an established, non-invasive medical imaging technique that produces detailed cross-sectional images of the body. Thus, low-dose CT scans can provide both 2D slices and 3D reconstructions of the human parietal bone, allowing for multiple analyses of the parietal skull, with details focusing on structure and thickness. In 2D analyses, the thickness of the parietal bone is measured by taking a series of measurements along a single plane of the CT image. This method can provide accurate measurements of bone thickness, but it is limited in its ability to capture the complex 3D structure of the parietal bone. While 2D analysis is particularly useful for identifying local thinning or thickening of the bone, 3D analyses allow for a more comprehensive view of the bone’s structure, shape, size, and thickness. Such 3D analyses can further confirm the 2D data if available. Furthermore, 3D mapping was particularly useful for identifying patterns of thinning or thickening across the entire bone (Figure 10). The 3D reconstruction was reliable, with average differences between the two models not exceeding 0.15 mm. However, CT-OAM and densitogram patterns do not provide qualitative data for direct comparisons.

For this work, advanced imaging software tools have been used according to their availability in this institution, and a commercial partner specialized in CAD and Mesh modeling for industrial purposes. However, these methods are not approved for medical purposes and were applied only for this pivotal study. Nevertheless, the proposed techniques provided data for the treating physician and suggested a possible effect of corticosteroids in this individual patient. Although this workflow from 2D to 3D analyses could not be validated in a more extensive case series, it provides first experiences which might be helpful even for interventional studies to monitor disease progression and evaluate the effectiveness of treatments.

Indeed, from a clinical perspective, the image analyses detailed above were considered helpful by the treating physicians. The stable disease under treatment with corticosteroids after day 229, which correlated with a clear improvement in headaches, suggested a beneficial effect of corticosteroids concerning parietal bone loss and excluded further deterioration of bone loss as a possible side effect of corticosteroids. Only quantifying parietal bone loss allowed for a comparison of the history of clinical symptoms such as headaches with the potential consequences of corticosteroid treatment. To our knowledge, Lichen ruber has not been described to result in bone changes similar to Paget’s bone disease. Although routine treatment of Lichen ruber does not include immunosuppressive agents such as corticosteroids, histology has defined Lichen ruber with inflammatory reactions around the hair follicles, including diffuse lymhocytic infiltrates. According to the anatomical proximity, a cytokine-mediated effect of such an inflammatory process on the near skull could not be excluded.

Nevertheless, the complete pathophysiological mechanisms involved are still unclear, as bone biopsy was not considered because of possible complications close to the central nervous system. Bone metastasis, which occurs in about 10% of cases of gastric cancer [23], was excluded as a differential diagnosis based on the clinical course and ^18^F-Fluorodexoyglucose positron emission tomography (FDG-PET). Thus, the clinical management aim in this aged patient was to control the symptoms and stop or even improve parietal bone loss. Fortunately, corticosteroids resulted in stable disease with fewer headaches.

Due to the low prevalence of parietal thinning, even case series are rare for follow-up analyses and have not been reported in the literature. This report compares routine descriptive reports with quantitative analyses using 2D and 3D evaluations of bone defects. This patient’s CTs showed increased bone loss independently from the methods applied between days 0 and 229. Indeed, a standard to quantify parietal bone loss cannot be derived from these data, and a multicentric study with more patients is necessary for the direct comparison of the proposed techniques. Any 2D and 3D evaluations provided helpful information for further treatment decisions.

Without doubt, the main limitation of this work is the lack of a greater number of identified patients with follow-up CTs of this rare disease—both with and without therapeutic interventions. More data will allow the further validation of the proposed analytical approaches, and it can be anticipated that 3D reconstructions and other image analyses will be helpful for diagnostic purposes and follow-ups. Specialized software approved for medical studies is needed for such investigations. Certainly, the results depend on the resolution of the CTs used. In this CT setting, at 0.6 mm, the nominal single collimation has to be considered as low.

## 5. Conclusions

In conclusion, for routine radiological practice, the use of the 3D reconstruction of CTs with subsequent 2D analyses appears fully justified to increase the clinical awareness of the size of parietal bone defects, allowing for the quantification of bone loss during long-term follow-up. Additional 3D analyses may then provide further information on patterns of bone thinning or thickening across the skull and allow for a more detailed follow-up of changes in bone volume over time. In view of these methods, the description of parietal thinning alone appears to be insufficient for the follow-up imaging of parietal thinning.

## Figures and Tables

**Figure 1 diagnostics-14-00446-f001:**
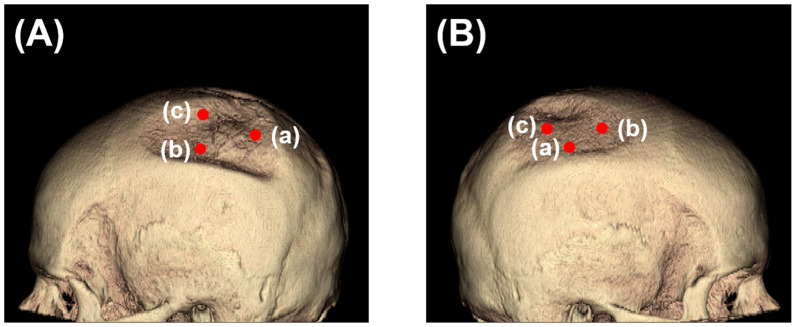
Selection of three sample points (**a**–**c**) on the left (**A**) and right (**B**) side for 2D measurements in 3D head models using CT images in the lateral views.

**Figure 2 diagnostics-14-00446-f002:**
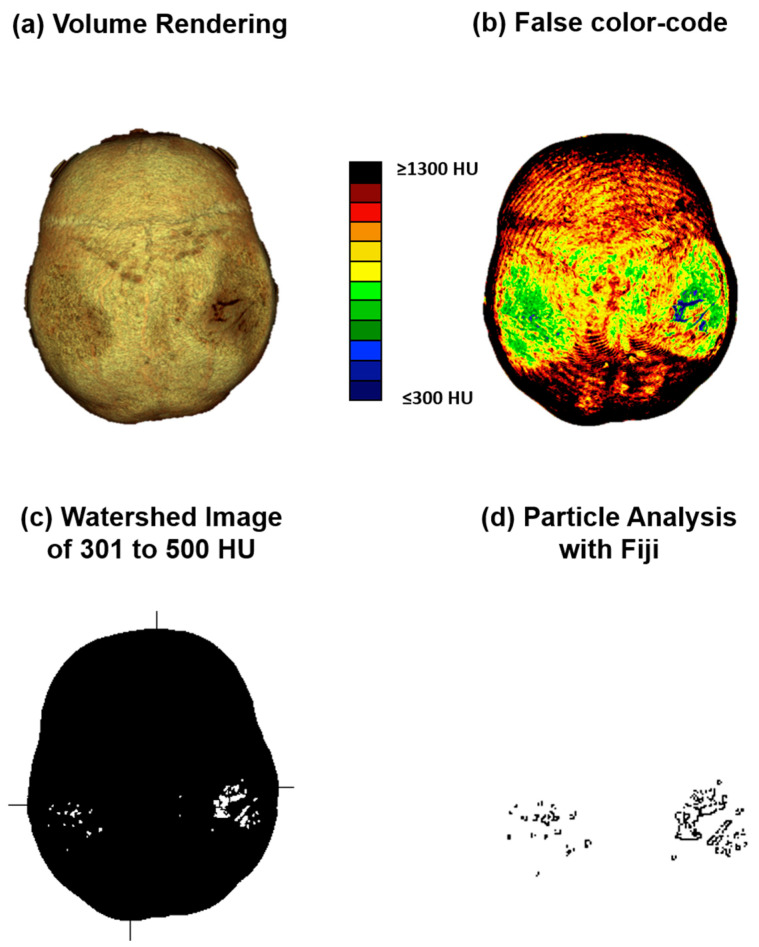
CT-OAM methodology. (**a**) CT data were uploaded to the image analysis software Analyze 14.0. The skulls were registered with 2D transformation using linear interpolation and reconstructed three-dimensionally. (**b**) Greyscale values corresponding to the respective HU were surface projected using a maximum intensity projection (MIP) algorithm generating a greyscale densitogram. A false color code corresponding to the HU units was applied. (**c**,**d**) The MIP color-code images were loaded into Fiji. The images were color split, threshold adapted, and watershed separated, and particles with values of 301 to 500 HU were analyzed.

**Figure 3 diagnostics-14-00446-f003:**
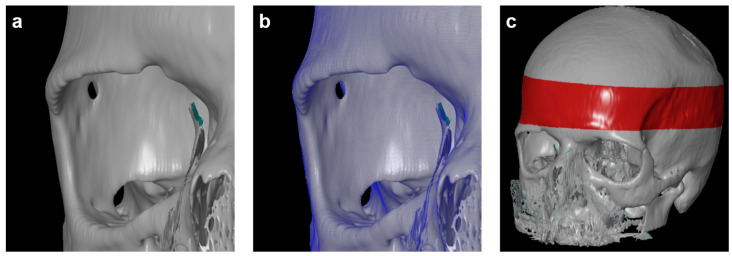
(**a**) Computer-aided design (CAD) model. (**b**) Overlay of CAD model with Mesh format. (**c**) Skull alignment. The 3D reconstruction is reliable, with average differences between the two models not exceeding 0.15 mm.

**Figure 4 diagnostics-14-00446-f004:**
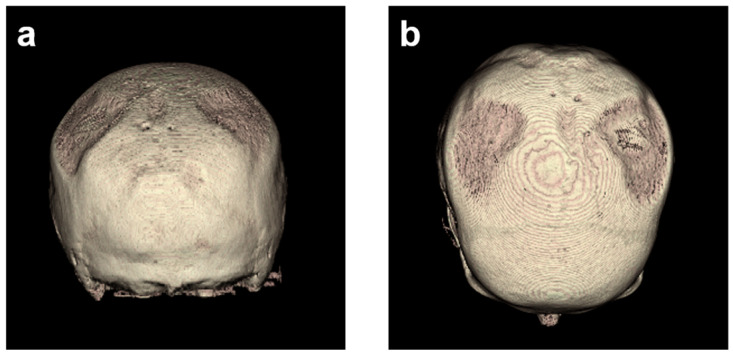
Three-dimensional head models based on the first CT images with reconstruction by semi-automatic object segmentation with region growth from a caudal (**a**) and dorsal view (**b**), displaying the parietal cranial bone. The corresponding lateral left and lateral right views are shown in Figure 1.

**Figure 5 diagnostics-14-00446-f005:**
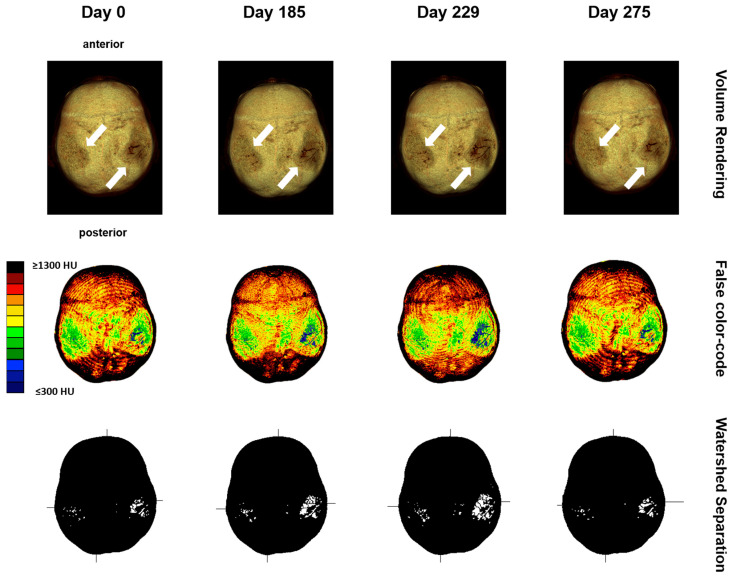
Maximum intensity projection with a false-color-code representation of values from 301 to 1300 HU of the skull bone showing progressive thinning on both parietal sides over time. Orientation is equalized for better comparison through registration using a 2D transformation with linear interpolation. Differences are marked with white arrows.

**Figure 6 diagnostics-14-00446-f006:**
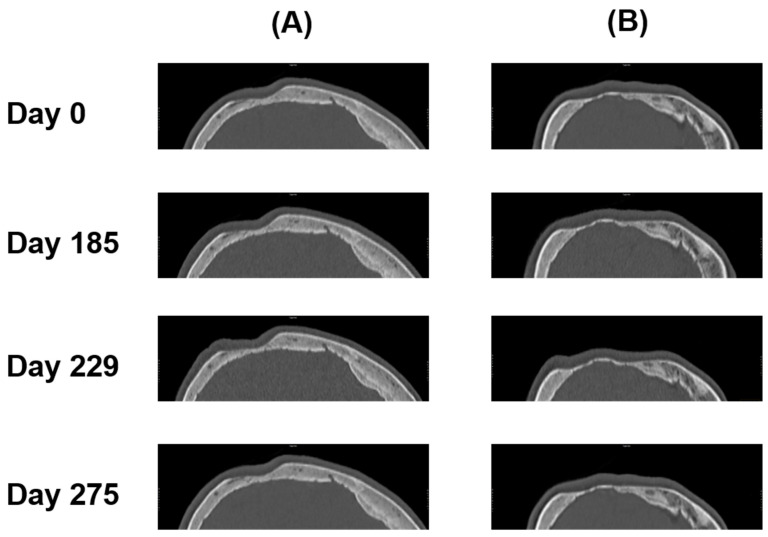
Sagittal CT images of a 75-year-old female patient on days 0, 185, 229, and 275 from the left (**A**) and the right (**B**) side.

**Figure 7 diagnostics-14-00446-f007:**
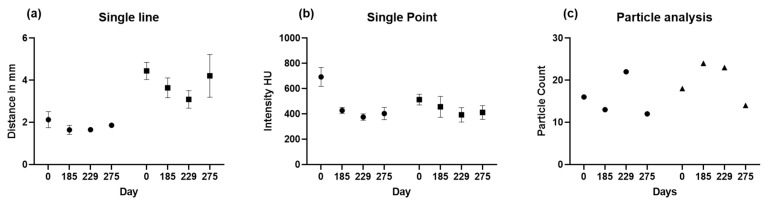
(**a**) Sample line (black circles = left side and black squares = right side), (**b**) sample point (black circles = left side and black squares = right side), and (**c**) particle analysis (black circles = left side and black triangles = right side) with 301 to 400 HU group summary plots are shown for the left and right side. The means with standard deviations are displayed for the sample line and sample point.

**Figure 8 diagnostics-14-00446-f008:**
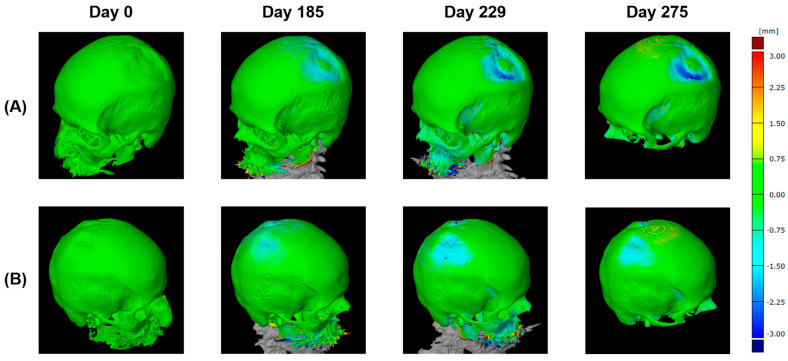
Two different perspectives of colored thickness mapping with defects up to −3mm between days 0 and 229/275 from the left (**A**) and the right (**B**) side.

**Figure 9 diagnostics-14-00446-f009:**
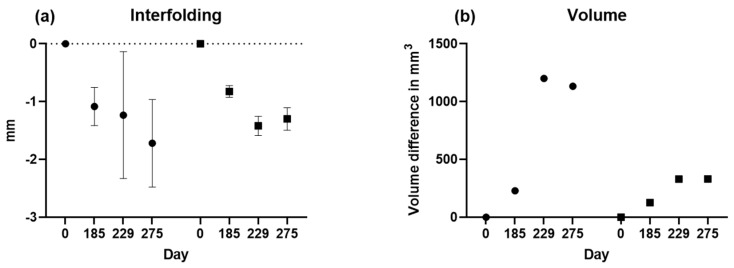
(**a**) Interfolding and (**b**) volume group summary plots are shown for the left and right sides. Means with standard deviations are displayed for the interfolding.

**Figure 10 diagnostics-14-00446-f010:**
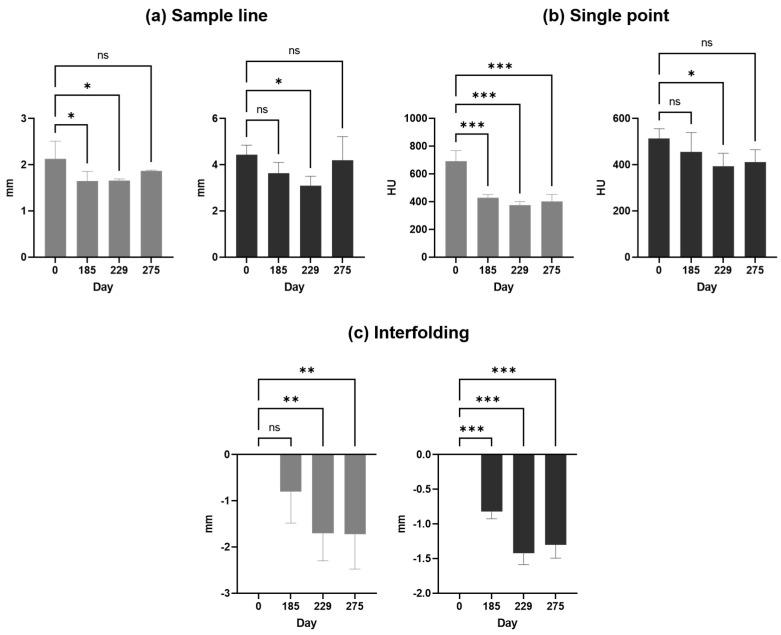
(**a**) Sample line, (**b**) single point, and (**c**) interfolding column bar graphs are shown for the left (grey) and right (black) sides as means with standard deviations. * Significant; ** high significant; *** highly significant difference between means; ns, not significant.

**Table 1 diagnostics-14-00446-t001:** Comparisons of bone loss were assessed by linear and volumetric measurements over time using the one-way ANOVA test and Fisher’s LSD test for multiple comparisons.

			*p*-Values Based on One-Way ANOVA			
Hemisphere	Name	Units	0 vs. 185	0 vs. 229	0 vs. 275	229 vs. 275
Right	Single line	mm	0.1559 ^ns^	0.0300 *	0.6581 ^ns^	0.0614 ^ns^
	Single point	HU	0.2840 ^ns^	0,0420 *	0,0755 ^ns^	0.7166 ^ns^
	Particle analysis	Counts	n.a.	n.a.	n.a.	n.a.
	Interfolding	mm	<0.0001 ***	<0.0001 ***	<0.0001 ***	0.3148^ns^
	Volume	mm	n.a.	n.a.	n.a.	n.a.
Left	Single line	mm	0.0263 *	0.0287 *	0.1747 ^ns^	0.2745 ^ns^
	Single point	HU	0.0001 ***	<0.0001 ***	<0.0001 ***	0.4904 ^ns^
	Particle analysis	Counts	n.a.	n.a.	n.a.	n.a.
	Interfolding	mm	0.1356 ^ns^	0.0075 ***	0.0072 ***	0.9786 ^ns^
	Volume	mm	n.a.	n.a.	n.a.	n.a.

*, significant; ***, highly significant difference between means; n.a., not assessable; ^ns^, not significant.

## Data Availability

The data presented in this study are available on request from the corresponding author. The data are not publicly available due to privacy reasons.

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
