# Peer review of "Long-Term Comparison of Two- and Three-Dimensional Computed Tomography Analyses of Cranial Bone Defects in Severe Parietal Thinning"

_diagnostics, 2024, doi:10.3390/diagnostics14040446_

Round 1

Reviewer 1 Report

Comments and Suggestions for Authors

The current study is submitted as a research paper; however, the structure is highly likely a case study. The study findings are based on a single case, which challenges the reproducibility of the study outcomes. The methods section still includes the text from the template, which highlights that you haven't read the final version of the article prior to submission. Additionally, there are not enough details presented for the analysis approaches and evaluation criteria. Please see the other comments below

1. Please correct typo in the title. 

2. Please revise the following statements of the abstract, " This study compares different methodological ap- 18 proaches including both two- and three-dimensional methods using the computerized tomogra- 19 phies of a single patient. The two-dimensional method provides accurate measurements of bone 20 thickness, while the three-dimensional method allows for a more comprehensive view of the bone's 21 structure, including parameters such as shape, size, and thickness. Both techniques provide valua- 22 ble data for the treating physician and suggest a possible effect of corticosteroids in this individual 23 patient."

3. The abstract is not specific for the current study. You're using very general terms to express the methods and results of the study. Also, no quantitative analysis at all. Please address these issues. 

4. Please revise and restructure the first paragraph of the introduction. You're jumping between descriptions, which makes it harder to focus on the topic. 

5. In the second paragraph, you should clearly state the need for the current study. Please mention the novelty of the current study. 

6. In Section 2.1, you're providing details for the case; however, this is a research article. Please revise it. 

7. Please provide details of the spatial resolution of the CT images. 

Comments on the Quality of English Language

The language is overall reads well. 

Author Response

Dear Editor,     Please find enclosed the revised version of the manuscript! Thank you for the valuable comments to improve the manuscript.   Kind regards,  

Comments to Reviewer 1:

The current study is submitted as a research paper; however, the structure is highly likely a case study. The study findings are based on a single case, which challenges the reproducibility of the study outcomes. The methods section still includes the text from the template, which highlights that you haven't read the final version of the article prior to submission. Additionally, there are not enough details presented for the analysis approaches and evaluation criteria. Please see the other comments below

Indeed it was difficult to decide the type of papereither as a research paper according to the comparison of different methods or a case report from the clinical perspective. We changed the format to a case report as proposed.

We excuse that we still included the text from the template, and deleted this text.

Thank you for the comments, we added the details for the analytic approaches and evaluation criteria as suggested in the comments below.

  1. Please correct typo in the title. We corrected the typo as proposed.

  1. Please revise the following statements of the abstract, " This study compares different methodological approaches including both two- and three-dimensional methods using the computerized tomographies of a single patient. The two-dimensional method provides accurate measurements of bone thickness, while the three-dimensional method allows for a more comprehensive view of the bone's structure, including parameters such as shape, size, and thickness. Both techniques provide valuable data for the treating physician and suggest a possible effect of corticosteroids in this individual patient."
  2. The abstract is not specific for the current study. You're using very general terms to express the methods and results of the study. Also, no quantitative analysis at all. Please address these issues.

Indeed, the quantitative analyses are more specific to describe the results in the abstract. The abstract was completely revised a ccording to the reviewe’s comments.

  1. Please revise and restructure the first paragraph of the introduction. You're jumping between descriptions, which makes it harder to focus on the topic.

The first paragraph of the introduction has been clarified, to allow better focusing on the different aspects of epidemiology and pathogenesis of parietal thinning.

  1. In the second paragraph, you should clearly state the need for the current study. Please mention the novelty of the current study.

This point is well taken and was added to the introduction.

  1. In Section 2.1, you're providing details for the case; however, this is a research article. Please revise it.

Indeed, as a case-report, the patient’s details are now presented in the result’s section.

  1. Please provide details of the spatial resolution of the CT images.

The nominal single collimation wifth is 0.6mm, the nominal total collimation with was 12mm. These data were added to the manuscript.

Reviewer 2 Report

Comments and Suggestions for Authors

This study is interesting as for clinical Relevance, however, I am not sure whether the research design was appropriate.

Since author indicated that three-dimensional reconstruction was followed by two-dimensional analyses. For clinical practice, if three-dimensional reconstruction is done, the skull defects could be easily show by 3D reconstruction. Therefore, I would recommend the author explain why this study is designed and how this 2D measurement is necessary if 3D reconstruction was already complete. 

Author Response

Dear Editor,     Please find enclosed the revised version of the manuscript! Thank you for the valuable comments to improve the manuscript.   Kind regards,

Comments to reviewer 2:

This study is interesting as for clinical Relevance, however, I am not sure whether the research design was appropriate.

Thank you for confirming the clinical relevance. We agree that this is a case-report as also mentioned by the other reviewer. We changed the type of the manuscript accordingly.

Since author indicated that three-dimensional reconstruction was followed by two-dimensional analyses. For clinical practice, if three-dimensional reconstruction is done, the skull defects could be easily show by 3D reconstruction. Therefore, I would recommend the author explain why this study is designed and how this 2D measurement is necessary if 3D reconstruction was already complete.

Indeed, the three-dimensional reconstruction was performed to further confirm the two-dimensional analyses. As already clarified according to the reviewer 1, the aim of the study was to provide full data for several two- and three-dimensional methods to quantify the bone defects in this case.

Round 2

Reviewer 1 Report

Comments and Suggestions for Authors

Thank you for revising the manuscript. I have minor comments. 

1. Please check citation guidelines with author names. 

2. Please check the software citation format of the journal. 

3. Please consider relocating the figure 2 to remove empty space. 

Author Response

Dear Reviewer 1,

On behalf of our coauthors, we thank you very much for allowing us to revise our manuscript.

We very much appreciate the constructive criticism. Please find our answer point by point below.

Yours sincerely

  1. Please check citation guidelines with author names.

Thank you for bringing this to our attention. We apologize for any oversight regarding the citation guidelines for author names. We reviewed the guidelines thoroughly and made necessary corrections.

  1. Please check the software citation format of the journal.

Thank you for bringing this to our attention. We apologize for any oversight regarding the citation guidelines for the software. We reviewed the guidelines thoroughly and made necessary corrections.

  1. Please consider relocating the figure 2 to remove empty space.

Thank you for your valuable feedback. We appreciate your suggestion to optimize the layout by relocating Figure 2 to remove empty space. We made the necessary adjustments to improve the overall presentation of Figur 2. If you have any further suggestions or concerns, please feel free to let us know.